# Trends in appropriateness of end-of-life care in people with cancer, COPD or with dementia measured with population-level quality indicators

**Robrecht De Schreye**[1]*, **Luc Deliens**[1,2], **Lieven Annemans**[3], **Birgit Gielen**[4], **Tinne Smets**[1ᵒ], **Joachim Cohen**[1ᵒ]

**1** End-of-life Care Research Group, Vrije Universiteit Brussel (VUB) and Ghent University, Brussels, Belgium, **2** Department of Medical Oncology, Ghent University, Ghent, Belgium, **3** Department of Public Health, Ghent University, Ghent, Belgium, **4** InterMutualistic Agency, Brussels, Belgium

ᵒ These authors contributed equally to this work.
* Robrecht.deschreye@sciensano.be

**Data Availability Statement:** Data cannot be shared publicly because of privacy protection reasons. Data are owned by the Intermutualistic

## Abstract

### Introduction

Measuring changes in the appropriateness of end-of-life care provided to patients with advanced illness such as cancer, COPD or dementia can help governments and practitioners improve service delivery and quality of life. However, an assessment of a possible shift in appropriateness of end-of-life care across the population is lacking.

### Aim

Measuring quality indicators with routinely collected population-level data, this study aims to evaluate the appropriateness of end-of-life care for people with cancer, COPD or dementia in Belgium.

### Design

A population-level decedent cohort study, using data from eight population-level databases, including death certificate and health claims data. We measured validated sets of quality indicators for appropriateness of end-of-life care.

### Setting/Participants

All people who died from cancer or COPD or with dementia between 1st January 2010 and 1st January 2016 in Belgium.

### Results

We identified three main trends over time across the three disease groups of increasing use of: family physicians in the last 30 days of life (+21.7% in cancer, +33.7% in COPD and +89.4% in dementia); specialist palliative care in the last 14 days of life (+4.6% in cancer,

Agency (https://www.ima-aim.be/), the Belgian Cancer Registry (http://kankerregister.org/) and Statistics Belgium (https://statbel.fgov.be/). The original datasets that were linked to create the dataset for this study are available from these institutions for researchers who meet the criteria for access to confidential data. Access to the original data can be requested by contacting director Birgit Gielen of the Intermutualistic Agency at birgit.gielen@aim-ima.be, the general office of the Belgian Cancer Registry at info@kankerregister. org and director Wendy Schelfhaut of Statistics Belgium at wendy.schelfaut@economie.fgov.be. The linked dataset created for this study cannot be shared for privacy protection reasons and will be destroyed in December 2021.

**Funding:** This study is part of a research study funded by the Research Foundation Flanders (FWO grant number G012414N). (website: https://www. fwo.be/en/) The funders had no role in study design, data collection and analysis, decision to publish, or preparation of the manuscript.

**Competing interests:** The authors have declared that no competing interests exist.

+36.9% in COPD, +17.8% in dementia); and emergency department in the last 30 days of life (+7.0% in cancer, +4.4% in COPD and +8.2% in dementia).

## Conclusions

Although we found an increase of both specialized palliative care and generalist palliative care use, we also found an increase in potentially inappropriate care, including ED and ICU admissions. To increase the quality of end-of-life care, both timely initiating (generalist and specialist) palliative care and avoiding potentially inappropriate care transitions, treatments and medications need to be quality performance targets.

## Introduction

In people with advanced serious diseases such as cancer, chronic obstructive pulmonary disease (COPD) or dementia, when death is imminent in the foreseeable future, the emphasis of care ideally gradually shifts to comfort care. Continuing disease-modifying treatment when the benefits no longer outweigh possible negative impacts in terms of quality of life, comfort or dignity, could be considered inappropriate end-of-life care [1, 2]. In contrast, appropriate care near the end of life, such as the timely initiation of palliative care, could increase well-being considerably in people suffering from advanced life-threatening conditions [3–6].

In the last decades, measuring the appropriateness of end-of-life care and changing care practices at the end of life towards more appropriate end-of-life care has been on the agenda of many researchers and policy makers [3, 5, 7–10]. In contrast, research shows there has mainly been an increase in aggressive end-of-life treatment in western countries. Pacetti et al., for instance, found high chemotherapy use during the last 4 weeks of life, with the individual clinician being the sole predictor of continuation of chemotherapy between 2010 and 2012 [11]. Teno et al. found an increase between 2000 and 2009 in the USA in intensive care unit (ICU) admission and healthcare transition in the final 30 days of life, despite an increase in out-of-hospital death and hospice use [12]. Similar research in the US, Canada and Korea shows an increasing occurrence of multiple emergency department (ED) visits, ICU admissions and hospital admissions in the last weeks of life [12–16].

Recent efforts from governments and practitioners have aimed to increase the appropriateness of end-of-life care and the quality of life of patients with advanced illness, for instance by promoting the timely introduction of palliative care and advance care planning [4, 17–19]. While traditionally, these efforts focus on people with cancer, efforts have also been made to increase access to palliative care and promote advance care planning for people with dementia and COPD [20, 21]. Appropriateness of end-of-life care received increasing attention from policy makers in countries such as the US, Canada, Germany and Belgium, among others [22–25]. When comparing Belgian government advisory documents of 2009 to those of 2017, the increasing emphasis on appropriateness of end-of-life care is evident, moving from providing definitions of palliative care in 2009 [26], to reporting on the appropriateness of end-of-life care and providing advice to promote palliative care use in 2017 [25]. Hence, we could expect a shift towards more appropriate end-of-life care.

While measuring and understanding changes in the quality of care provided to patients at the end-of-life at a population level through appropriateness and inappropriateness indicators can help governments and practitioners improve service delivery and patients' quality of life, a comprehensive assessment of the appropriateness of end-of-life care across the population,

measuring a wide range of indicators of both appropriate and inappropriate end-of-life care is currently lacking. Using a previously developed quality indicators set based on administrative data [27], this study aims to evaluate the appropriateness and inappropriateness of end-of-life care for people with cancer, COPD or dementia between 2010 and 2015 in Belgium. The specific research questions are:

1. Did the appropriateness of end-of-life care for people dying from cancer or COPD or with dementia change between 2010 and 2015 in Belgium?

2. What indicators of appropriate or inappropriate end-of-life care have had the largest changes in this time period?

## Methods

### Study design and data sources

We conducted a population-level decedent cohort study of all deceased from cancer, from COPD or with dementia between 1st January 2010 and 1st January 2016 in Belgium. Cancer, COPD and dementia represent the three distinct disease trajectories for people with chronic progressive illness, identified by Murray et al. [28] Data from eight administratively managed population-level databases was linked [29]. At the moment of the linking procedure, data from 2015 were the most recent data fully available in all linked databases. As healthcare insurance is legally mandatory in Belgium, the central health care claims databases contain data on almost the entire Belgian population [30].

(1) The health care claims database containing all health care use data of reimbursed home, nursing home and hospital care, except medication dispensed in public pharmacies;

(2) The pharmaceutical database containing data on all reimbursed medication dispensed in public pharmacies;

(3) The database with socio-demographic data on all people with healthcare insurance;

(4) The Cancer Registry database with data on all incidences of cancer including the type of cancer and date of diagnosis of each incidence;

(5) The national death certificate database containing all registered causes of death;

(6) The population registry database with data on household composition;

(7) The census database, containing data from national censuses held in 2011 and 2001, including housing characteristics and educational level;

(8) The fiscal database, including the net taxable income of each Belgian citizen;

After acquiring approval from all relevant data protection agencies, all databases were linked in a secure and ethically responsible manner, guaranteeing anonymity of the deceased [29]. The database linking procedure is described in detail in a previous publication [29].

### Population

We included all people who died from cancer or COPD or with dementia between 1st January 2010 and 1st January 2016 in Belgium. People dying from cancer (ICD10 codes C00-C99) or COPD (ICD10 codes J41-J44) were identified using the underlying cause of death on the death certificate. People who died with dementia were identified using the underlying, intermediate, external and associated causes of death reported on the death certificate (ICD-10 codes F00, F01, F02, F03 or G30). We used this broader selection because dementia is often underreported as a primary cause of death and people with advanced dementia often die from other causes, most commonly pneumonia [31, 32]. That is why, in contrast to cancer and COPD, we report on people dying 'with' and not necessarily 'from' dementia.

## Data

In the Belgian health care system, health care costs for a wide range of treatments and medication are reimbursed to the patient. Every reimbursement is registered by health insurers, while a central agency collects data from all health insurance registries [30]. The resulting central health and pharmacy databases were part of the linking procedure mentioned in the methods section [29]. From these databases, we used all available data on health care use including treatment, medication use, dates of treatment and prescription and admission to hospitals, emergency departments, intensive care units and nursing homes. We also selected multiple socio-economic, demographic and clinical variables from other administrative databases that might influence health care use including age, gender, net taxable income, dependence on care, highest attained level of education, household composition and degree of urbanization of the municipality of residence.

## Quality indicators

We used three validated sets of quality indicators for appropriateness of end-of-life care: one for people dying with dementia (28 indicators), one for people dying from COPD (28 indicators) and one for people dying from cancer (26 indicators). They were developed and validated using a RAND/UCLA Appropriateness method [9, 33]. The expert panels for this validation consisted of general practitioners, pharmacologists and palliative care specialists, adding neurologists, nursing home coordinating and advisory physicians and geriatrists for indicators for people with dementia, oncologists, pneumologists and radiotherapists for indicators for people with cancer and pneumologists for indicators for people with COPD [9].

The quality indicators measure the prevalence of specific healthcare interventions in a specified period before death (e.g. 7 days, 14 days, 30 days, 180 days), within the specific population they were developed for. For the purpose of this trend study all indicators were measured for the period of 30 days before death, except when specific indicators were not validated by the expert panel for that time period, or time before death was irrelevant for the quality indicator. One indicator measuring tube feeding or intravenous feeding could not be measured with the available data, as these are not reimbursed as individual interventions, but as a package in the hospital context.

Being population-level quality indicators, these indicators are not used to assess the quality of care of individual patients, nor do they imply the aim to be reduced to 0% (for inappropriate care) or 100% (for appropriate care).

## Risk adjustment and trend analysis

To obtain a trend analysis portraying real evolutions in appropriateness of end-of-life care rather than changes in the risk profile of the population (i.e. attributional validity [34]), we controlled for variables that changed across years and might influence health care use. We used a comprehensive list of possible confounders and made a selection based on availability in the linked dataset [29]. The following risk factors were identified as relevant and measurable with the current dataset: age, gender, highest attained level of education, net taxable income, household composition (e.g. married, single, with or without children, in a nursing home), degree of urbanization of the municipality of residence, being officially recognized as having high care needs, and being entitled to a higher degree of reimbursement due to lower degree of self-reliance. These last two variables represent measures of care dependence, which is known to be linked with multimorbidity [35], and they are based on receiving any of the allowances for a highly care-dependent person as judged by the GP. These variables were used to calculate risk adjusted quality indicator scores for each of the years.

## Statistical analyses

We calculated the population characteristics using descriptive statistics. To perform the risk adjusted trend analysis, we used stepwise logistic regression model building (with a significance level of 0.3 for entry and 0.05 to stay in the model), to identify what variables were associated with each quality indicator (coded binary as present [1] or absent [0]) across years. Using the results from this predictive model, we calculated a predicted quality indicator score for each quality indicator per year. A risk-adjusted score was then calculated by dividing the average predicted score by the average observed score for each year, multiplied by the average observed score across all years. This way, a trend analysis was performed, taking into account relevant population differences across years.

We also calculated all risk adjusted indicator score differences between 2010 and 2015. We subtracted the indicator scores in 2015 with the indicator scores of 2010 and divided this by the score in 2010, to obtain a relative (percentage) increase. We then inverted all results of indicators of inappropriateness of care (multiplied by -1) and joined all indicators for each disease (cancer, COPD, dementia) in a ranking from most positive to most negative evolutions.

All analyses were conducted with SAS Enterprise Guide, version 7.1. Programming codes are available from the authors on request.

## Reporting and ethics

We report our results following the STROBE and RECORD guidelines for observational routinely-collected health data [36].

The study was approved by the Brussels university hospital committee for medical ethics (B.U.N. 143201627075).

The administrative data linking process was approved by the national Data Protection Authority (project SA1/STAT/MA-2015-026-020-MAV) and by the Statistical Monitoring Committee (project STAT-MA-2015-026). Only anonymized data were used and small cell analysis was performed to prevent re-identification.

## Data sharing

Unfortunately, the data are not publicly available and due to privacy regulations, cannot be shared by the authors.

## Results

### Cohort characteristics

634,445 people died between 1st January 2010 and 1st January 2016 in Belgium, of which 25.2% (159,590) died from cancer, 6.0% (37,930) died from COPD and 9.5% (59,967) with dementia (Table 1). The mean age at death in 2010 was 68.2 in people with cancer, 72.0 in people with COPD and 79.5 in people with dementia. The mean age at death in 2015 was 73.8 in people with cancer, 78.2 in people with COPD and 86.6 in people with dementia.

### Risk adjusted differences in quality indicators scores between 2010 and 2015 (Table 2)

Across all three disease groups, we found a large increase between 2010 and 2015 in the percentage of people who had an increased contact with a family physician towards the end of life, i.e. more contacts with the GP in the final 30 days than in the period before (+21.7% in cancer, +33.7% in COPD and +89.4% in dementia). We also found an overall increase in people who

**Table 1. Summarized population description of all people who died from cancer, with COPD or with dementia in Belgium from 2010 until 2015[a].**

| | | Dying from Cancer (N = 159,590) | | Dying with COPD (N = 37,930) | | Dying with Dementia (N = 59,967) | |
|---|---|---|---|---|---|---|---|
| | | 2010 (N = 26,768) | 2015 (N = 26,493) | 2010 (N = 6,878) | 2015 (N = 6,185) | 2010 (N = 10,017) | 2015 (N = 10,629) |
| **Age mean (SD)[b]** | | 68.2 (7.2) | 73.8 (7.5) | 72.0 (7.8) | 78.2 (7.6) | 79.5 (8.9) | 86.6 (8.6) |
| **Agecategory** | <65 | 24.6 | 22.9 | 11.7 | 11.6 | 1.3 | 0.8 |
| | 65–74 | 23.6 | 24.3 | 19.4 | 21.2 | 5.1 | 3.9 |
| | 75–84 | 35.6 | 34.4 | 44.5 | 40.6 | 41.5 | 34.3 |
| | >84 | 16.2 | 18.4 | 24.4 | 26.6 | 52.1 | 61.1 |
| **Sex** | Female | 43.6 | 44.1 | 32.7 | 36.7 | 65.4 | 65.8 |
| **Householdtype** | Single | 29.3 | 29.9 | 31.6 | 31.3 | 20.4 | 18.8 |
| | Single parent | 5.0 | 5.3 | 5.2 | 4.7 | 3.8 | 3.7 |
| | Couple with children | 12.1 | 11.5 | 7.6 | 7.1 | 4.4 | 3.9 |
| | Couple without children | 44.7 | 44.4 | 37.2 | 36.5 | 27.2 | 24.9 |
| | Collective (i.e. nursing home) | 7.1 | 7.2 | 15.9 | 18.1 | 41.9 | 46.5 |
| | Other | 1.9 | 1.7 | 2.5 | 2.2 | 2.4 | 2.2 |
| **Housing Comfort** | High | 47.2 | 51.6 | 35.8 | 39.8 | 34.5 | 40.0 |
| | Average | 17.1 | 14.9 | 20.6 | 18.7 | 27.4 | 23.4 |
| | Low | 26.9 | 26.3 | 30.5 | 31.3 | 25.8 | 26.5 |
| | None | 8.8 | 7.2 | 13.1 | 10.2 | 12.3 | 10.2 |
| **Degree of urbanization of residence** | Very high | 31.5 | 30.0 | 32.6 | 30.9 | 33.9 | 30.9 |
| | High | 28.2 | 28.8 | 26.3 | 27.5 | 28.6 | 28.8 |
| | Average | 26.7 | 26.2 | 25.7 | 24.3 | 24.9 | 25.4 |
| | Low | 13.6 | 13.9 | 15.3 | 16.2 | 12.6 | 13.2 |
| **Region** | Flanders | 58.9 | 58.3 | 49.6 | 48.1 | 57.9 | 57.0 |
| | Wallonia | 33.1 | 34.0 | 41.7 | 44.5 | 32.3 | 35.3 |
| | Brussels | 8.0 | 7.7 | 8.7 | 7.4 | 9.8 | 7.7 |
| **Cancertype** | Respiratory | 27.8 | 25.2 | | | | |
| | Digestive tract | 28.9 | 28.4 | | | | |
| | Urinary tract | 6.4 | 6.5 | | | | |
| | Head and neck | 3.2 | 3.4 | | | | |
| | Melanoma | 2.6 | 3.4 | | | | |
| | Breast | 6.9 | 7.6 | | | | |
| | Female genital organs | 5.2 | 4.9 | | | | |
| | Male genital organs | 4.9 | 5.9 | | | | |
| | Other | 13.9 | 14.7 | | | | |

[a]Columns represent column percentages, except for the average age. All missing values are below 10%

[b]A detailed description of all population variables across all years can be found as S1–S3 Tables.

received specialized palliative care, (+5.4% in cancer, +25.6% in COPD, +8.8% in dementia) with particularly a late initiation of palliative care (in last 14 days only) increasing, (+4.6% in cancer, +36.9% in COPD, +17.8% in dementia). Another major trend across all three disease groups is the increase in percentage of people who were admitted to an emergency department in the last 30 days (+7.0% in cancer, +4.4% in COPD and +8.2% in people dying with dementia) (Table 2, Figs 1–3).

**Table 2. Trends overview, containing all risk adjusted differences between 2010 and 2015, ranking all indicators[a] from most positive to most negative evolution in relative percentage increase[b].**

| Cancer indicators of appropriate end-of-life care | Improvement score[b] | | COPD indicators of appropriate end-of-life care | Improvement score[b] | | Dementia indicators of appropriate end-of-life care | Improvement score[b] | |
|---|---|---|---|---|---|---|---|---|
| | Relative increase in indicator score between 2010 and 2015 | Absolute increase in indicator score between 2010 and 2015 | | Relative increase in indicator score between 2010 and 2015 | Absolute increase in indicator score between 2010 and 2015 | | Relative increase in indicator score between 2010 and 2015 | Absolute increase in indicator score between 2010 and 2015 |
| Increased number of contact with family physician | +21.7% | +9.8 | Increased number of contact with family physician | +33.7% | +13.7 | Increased number of contact with family physician | +89.4% | +32.5 |
| Multidisciplinary oncology consult | +14.7% | +1.8 | Specialist palliative care | +25.6% | +2.6 | Opioids and neuropathic medication | +24.5% | +0.4 |
| Specialist palliative care | +5.4% | +2.5 | Official palliative care status | +23.7% | +1.9 | Official palliative care status | +10.7% | +0.6 |
| Official palliative care status | +3.8% | +1.4 | Opioids | +6.0% | +2.3 | Specialist palliative care | +8.8% | +0.8 |
| Opioids and neuropathic medication | +2.6% | +0.2 | Death at home or in nursing home where resided for at least 180 days | +4.5% | +2.0 | Death at home or in nursing home where resided for at least 180 days | +1.6% | +1.1 |
| Opioids | +1.0% | +0.8 | Death at home | +1.0% | +0.3 | Death at home | -16.1% | -2.0 |
| Death at home or in nursing home where resided for at least 180 days | +0.1% | +0.1 | | | | | | |
| Death at home | -1.9% | -0.6 | | | | | | |
| Cancer indicators of inappropriate end-of-life care | Improvement score[b] | | COPD indicators of inappropriate end-of-life care | Improvement score[b] | | Dementia indicators of inappropriate end-of-life care | Improvement score[b] | |
| ICU admissions from a nursing home | -9.3% | -0.2 | Start taking an antidepressant | -20.4% | -1.1 | Blood transfusion (last 14 days before death) | -26.2% | -0.1 |
| Chemotherapy | -5.8% | -1.0 | Continuous endotracheal intubation | -20.2% | -1.4 | ICU admissions from nursing home | -19.1% | -0.2 |
| Diagnostic testing—ECG or pulmonary function testing | -0.5% | -0.2 | Reanimation after intubation | -17.0% | -0.3 | ICU admissions | -17.1% | -0.5 |
| Diagnostic testing (all) | -0.2% | -0.1 | Coronary or abdominal surgery | -16.7% | -0.2 | Dispension of serotonin reuptake inhibitors | -11.4% | -1.3 |
| Diagnostic testing—medical imaging | +0.1% | +0.0 | Blood transfusion (last 14 days before death) | -16.2% | -0.2 | Diagnostic testing—ECG or pulmonary function testing | -3.9% | -0.9 |
| Start taking an antidepressant | +0.1% | +0.0 | Repeated endotracheal intubation | -8.0% | -1.2 | Hospital death | -3.2% | -0.8% |
| Blood transfusion (last 14 days before death) | +0.7% | +0.0 | Endotracheal intubation or tracheotomy | -7.1% | -1.2 | Hospital admissions | -0.1% | -0.0 |
| Hospital admissions | +0.8% | +0.5 | Diagnostic testing—ECG or pulmonary function testing | -4.8% | -2.1 | Diagnostic testing (all) | +0.1% | +0.0 |

*(Continued)*

**Table 2.** (Continued)

| | | | | | | | | |
|---|---|---|---|---|---|---|---|---|
| Surgery | +1.0% | +0.0 | Hospital death | -3.9% | -2.1 | Diagnostic testing—medical imaging | +0.7% | +0.2 |
| Feeding tube or intravenous feeding | +3.7% | +0.1 | ICU admissions from nursing home | -1.7% | -0.0 | Dispension of anti-hypertensives | +4.5% | +1.9 |
| Late initiation of palliative care | +4.6% | +1.6 | Diagnostic testing (all) | -0.3% | +0.2 | ED admissions | +8.2% | +1.9 |
| ED admissions | +7.0% | +2.3 | Inhalation therapy | -0.2% | -0.2 | Dispension of NOAC's or vitamin K antagonists | +9.0% | +3.2 |
| | | | Diagnostic testing—medical imaging | +0.1% | +0.0 | Dispension of prophylactic gout medication | +13.4% | +0.2 |
| | | | Hospital admissions | +1.7% | +1.0 | Dispension of statins | +14.5% | +0.8 |
| | | | Late initiation of physiotherapy | +3.1% | +0.2 | Late initiation of palliative care | +17.8% | +0.7 |
| | | | Surgery | +8.3% | +0.1 | Chemotherapy | +23.5% | +0.1 |
| | | | ED admissions | +10.9% | +4.4 | Dispension of gastric protectors | +23.7% | +5.0 |
| | | | ICU admissions | +12.3% | +1.6 | Surgery | +32.5% | +0.2 |
| | | | Late initiation of palliative care | +36.9% | +1.3 | Dispension of calcium vitamin D | +36.0% | +2.3 |

[a]Only indicators measured as a percentage are included in this table, to enable comparability.

[b]For the ranking order, all results of indicators of inappropriateness of care were inverted (multiplied by -1), so a higher ranking means a decrease of these indicators

**Cancer.** For those who died from cancer, we additionally found an increasing percentage receiving a multidisciplinary oncology consult from 12.3% to 14.1% and a decrease of those receiving chemotherapy in the last 30 days of life from 17.5% to 16.5% (Table 2, Fig 1).

**COPD.** For COPD, we found an increase in the percentage of people receiving the official palliative care status (+23.7%) and the percentage of people who received opioids (+6.0%). We also found a decrease in the percentage of people receiving endotracheal intubation or tracheotomy (-7.12%), continuous (-20.18%) and repeated endotracheal intubation (-7.95%). However, the percentage of people admitted to the ICU increased (+12.29%) (Table 2, Fig 2).

**Dementia.** In people who died with dementia, in addition to the changes described above, we found an increase in the percentage of people with dispension of Novel Oral Anticoagulants (NOAC's) or vitamin K antagonists (+8.98%), gastric protectors (+23.67%), and calcium and vitamin D (+36.04%). The percentage receiving serotonin reuptake inhibitors decreased (11.44%) (Table 2, Fig 3).

## Discussion

### Main findings

Using full-population data, we observed three trends in the appropriateness and inappropriateness of end-of-life care that were common across cancer, COPD and dementia decedents: (1) a large increase in the percentage of people who had an increased contact with a family physician in the last 30 days, (2) an increase in the proportion who received specialized palliative care (mostly an increased late initiation of palliative care, i.e. in the last 14 days), and (3) a large increase in the percentage of people admitted to an emergency department in the last 30 days. Additionally, we found several trends specifically for people dying from cancer, from COPD or with dementia.

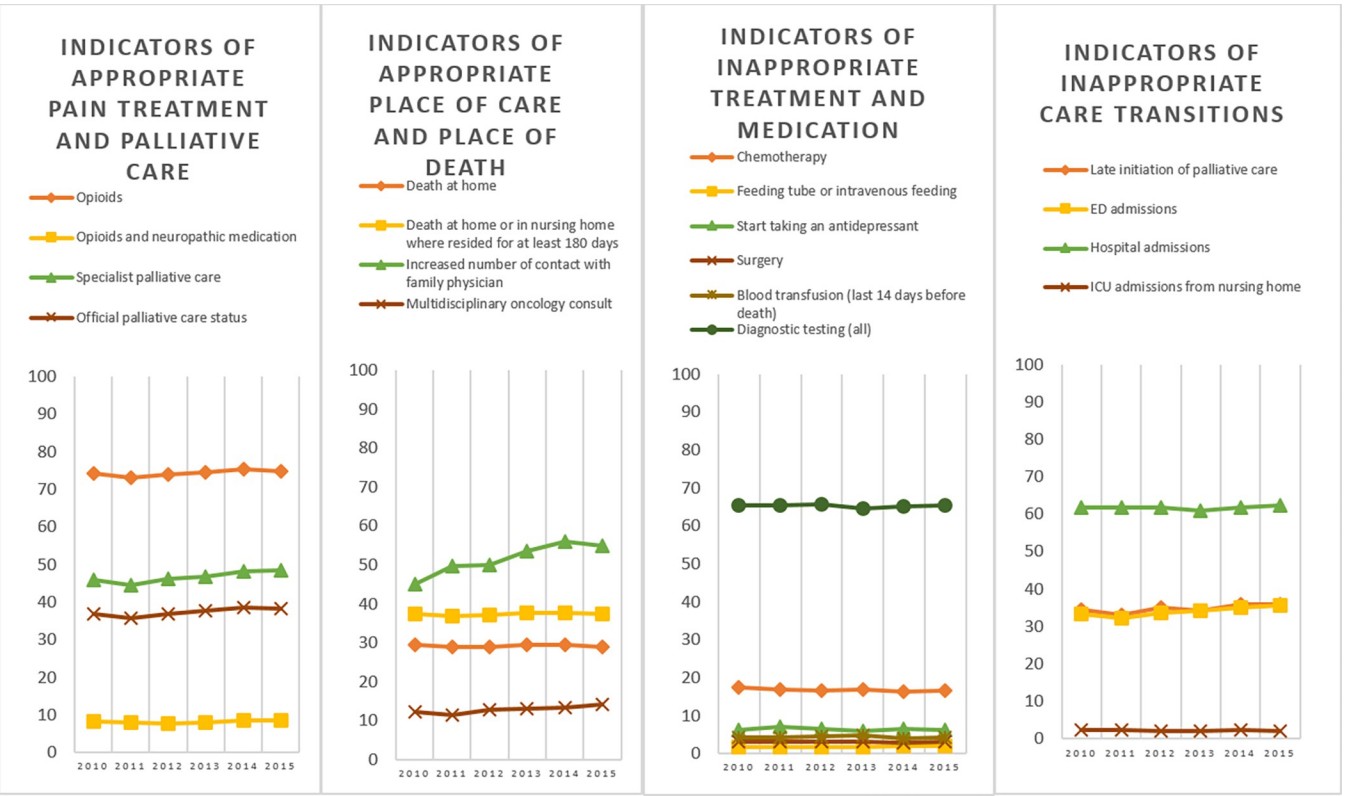

**Fig 1. Trends in indicators of appropriate or inappropriate care in the last 30 days of life for people dying from cancer, thematically grouped.** The Y-axis represents the risk adjusted percentage of the population that received the treatment or care specified for this quality indicator.

### Strengths and limitations

A first strength of this study is that the use of routinely collected administrative data allows us to study diverse and hard-to-reach populations, such as cognitively impaired and vulnerable participants, without discrimination. Other research suffers from issues adapting data collection methods for these populations (e.g. translating or adapting questionnaires) or selection bias, where our data collection method includes (almost) the entire Belgian population [36–38]. Including people with COPD and dementia is a second strength of this study, as end-of-life care research has traditionally mainly focused on people with cancer [39, 40]. Linking administrative databases including socio-demographic and death certificate databases also provides information on possible confounding variables, so we can observe trends in health care provision, unaffected by possible demographic and diagnostic evolutions. Finally, the data represent the full Belgian population in 2010–2015 and therefore provide generalizable information on the entire population's health care use.

A first limitation of this study is that we rely on death certificate data to select the COPD and dementia population. Literature suggests death certificate data tend to underestimate the prevalence of dementia [41, 42]. A second limitation is that the selected routinely collected databases do not contain data on non-reimbursed care. Some important aspects of care, such as patient-carer communication, patients' preferences, or health care that is not specifically reimbursed, such as palliative care included in regular nursing home care, cannot be measured. This implies that important aspects of appropriateness of end-of-life care are not measured in this study, but also that there might be residual confounding in comparing quality

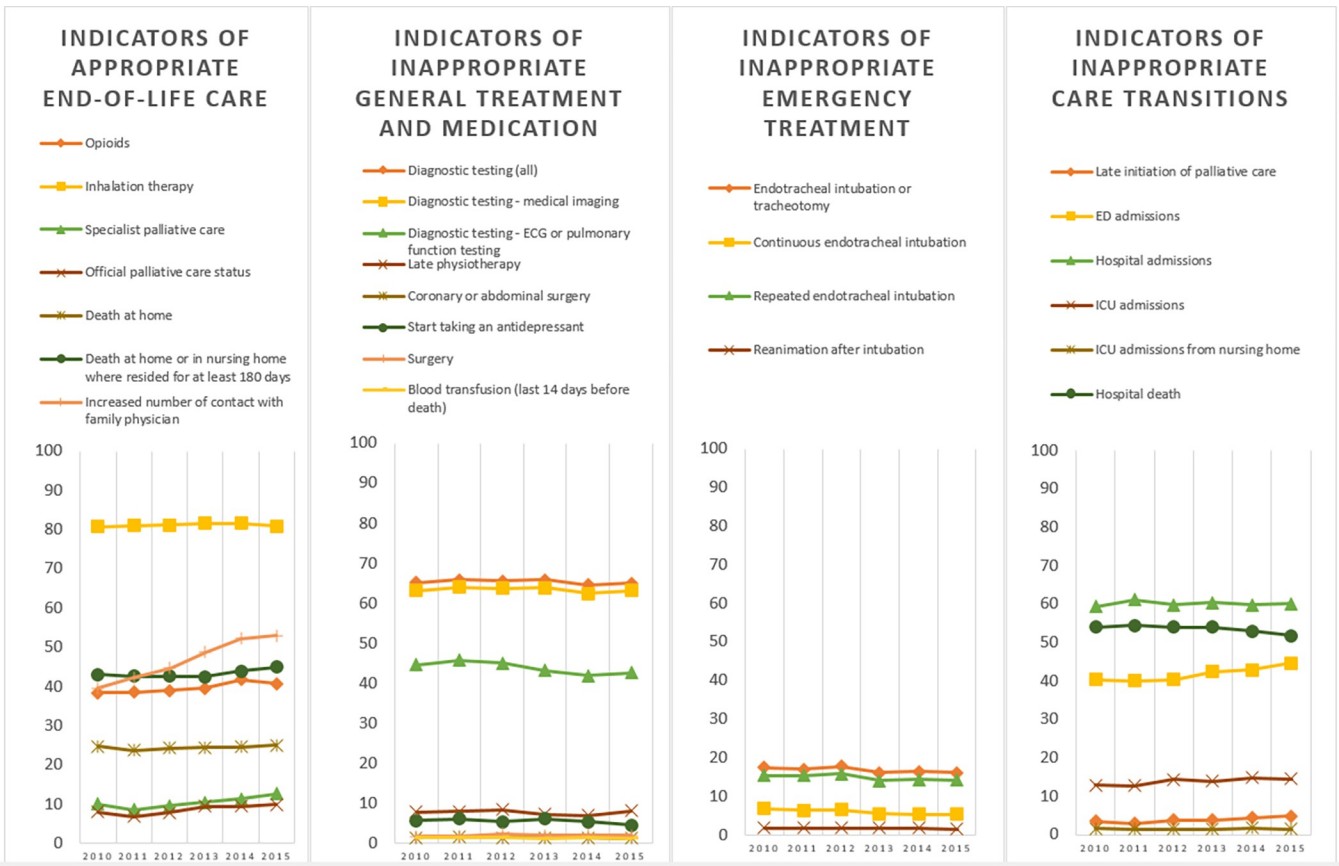

**Fig 2. Trends in indicators of appropriate or inappropriate care in the last 30 days of life for people for people dying from COPD, thematically grouped.**
The Y-axis represents the risk adjusted percentage of the population that received the treatment or care specified for this quality indicator.

indicator scores across years. However, we assume to have captured the most impactful population-level changes between years with available routinely collected data.

## Interpretation

We found an overall increase in the percentage of people receiving specialized palliative care, with the largest growth for people dying from COPD. This seems to imply the increasing emphasis on appropriateness of end-of-life care in Belgian government advisory documents between 2009 and 2017 [25, 26] coincides with an increased use of specialized palliative care. This increase, however, seems mostly due to specialized palliative care initiated in the last two weeks of life. A remaining challenge would, therefore, be to match increased access to palliative care service with their increased timely initiation. The percentage of people with increasing contacts with a general practitioner (GP) near the end of life has also risen. This might entail that so-called generalist palliative care, in which the GP usually plays a large role [43], is also increasing. General evolutions in the organization of primary care and the increased use of electronic health records in primary care might also have contributed to this increase, though further research is required to determine the contribution of these factor.

Despite the increase in palliative care, we found an increase in several indicators of possible inappropriate end-of-life care, most notably emergency department (ED) visits rising in all disease groups and ICU admissions in people dying from COPD or with dementia. These

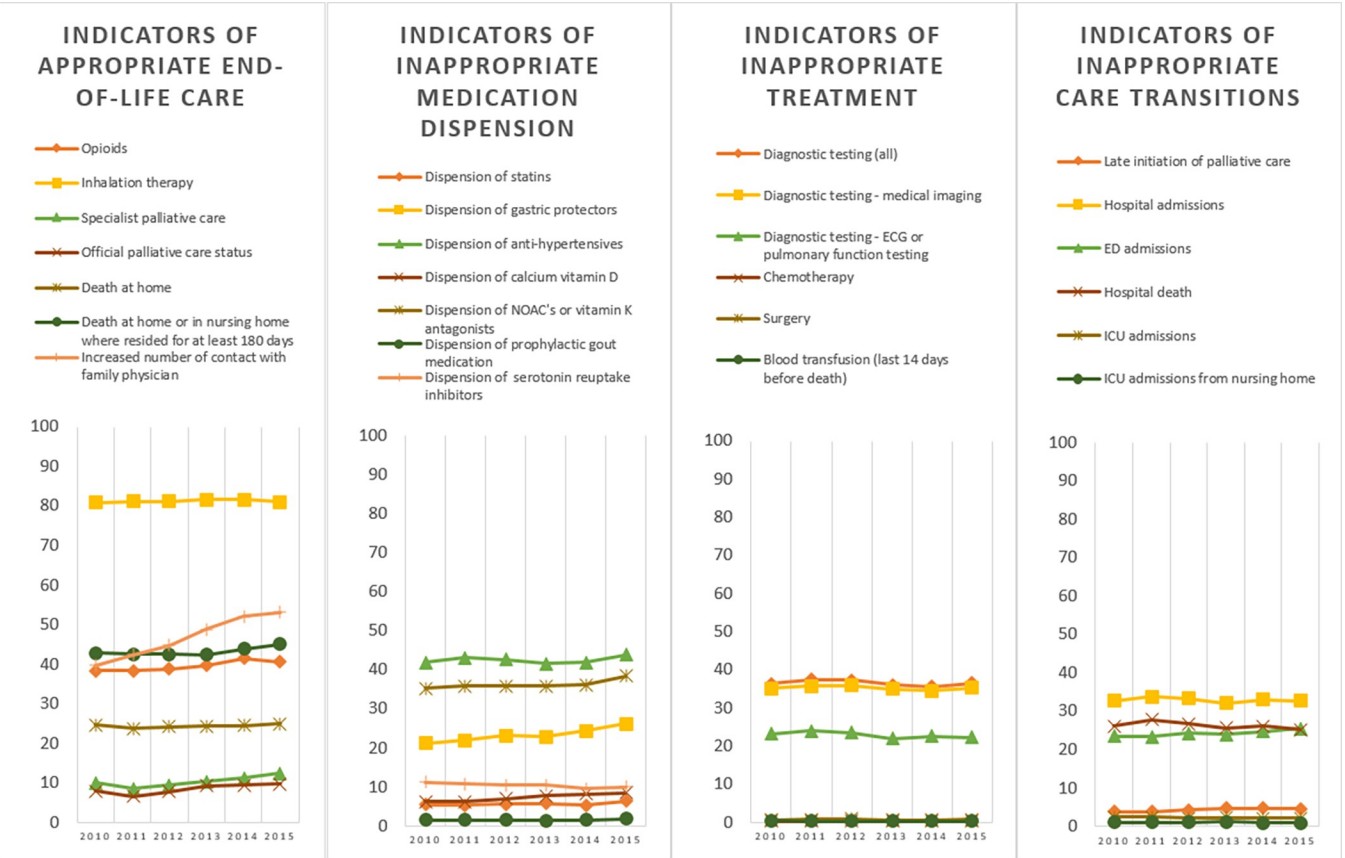

**Fig 3. Trends in indicators of appropriate or inappropriate care in the last 30 days of life for people for people dying with dementia, thematically grouped.** The Y-axis represents the risk adjusted percentage of the population that received the treatment or care specified for this quality indicator.

seemingly contradictory findings are in line with the results of Teno et al. [12], who found an increased use over time in the USA of hospice and hospital-based palliative care near the end of life, but at the same time a reduced average length of stay in the hospice, alongside an increase in ICU admissions and multiple hospital visits near the end of life. Between 2010 and 2015, no relevant policy changes that could explain this evolution were observed. With Teno et al. we can argue that increased use of palliative care does not reduce resource use in general. This suggests that palliative care is often combined with aggressive treatment rather than replacing it [12, 44].

For people dying with dementia, we found an alarming trend in medication provision: only the percentage of people receiving serotonine reuptake inhibitors in the last month of life is decreasing, while rates for all other indicators on possibly inappropriate medication are rising. It has been argued that polypharmacy and the continued intake of long-term preventive medication has adverse effects on the health and quality of life of frail older adults, including those with dementia [8, 45]. People with dementia near the end of life in Belgium often live in a nursing home. Studies report a number of barriers to discontinuing medication in this context, including perceived opposition of the patient's family or the patient him/herself [46, 47]. It has been argued that the initiation of some of the medications mentioned might benefit patients in a palliative care context, to treat symptoms [47]. This, coupled with a lack of discontinuation, would lead to a rise in medication use. However, the use of standardized medication reviews with tools such as the STOPPFrail [45] criteria, which have been circulated from 2015

onwards, could facilitate the discontinuation of inappropriate medication prescription in people with dementia.

## Implications

Overall, our findings suggest there is an increase in several indicators of appropriate end-of-life care, but little reduction in indicators of inappropriateness of end-of-life care. The increased use of specialized palliative care does not come with a population-level reduction in inappropriate end-of-life care.

We suggest that, to increase appropriateness of end-of-life care, an effort should be made to look for the causes of ED and ICU admissions of people with serious and chronic illness near the end of life, to address these causes, and to increase the availability of community alternatives such as palliative home care available to meet these people's complex care needs. Palliative care should be initiated earlier and, especially for people with dementia, medication policies should focus on deprescribing and discontinuation of inappropriate medication. Systematic monitoring of quality indicator scores and reestablishing relative standards, could help define goals for future improvement of the quality of end-of-life care [48, 49].

## Conclusion

We present a trend analysis of a large set of validated indicators of appropriate and inappropriate end-of-life care in people dying from cancer, from COPD or with dementia. Although we found indications of an increase of both specialized palliative care use and generalist palliative care use, we also found an increase in several indicators of inappropriate care in the last month of life, including ED and ICU admissions. To increase the appropriateness of end-of-life care, policy makers and practitioners should focus both on timely initiation of palliative care and on avoiding or discontinuing potentially inappropriate care transitions, treatments and medication.

## Supporting information

**S1 Table. Overview of all measured population characteristics of people dying from cancer in Belgium, from 2010 until 2015.**
(DOCX)

**S2 Table. Overview of all measured population characteristics of people dying from COPD in Belgium, from 2010 until 2015.**
(DOCX)

**S3 Table. Overview of all measured population characteristics of people dying with dementia in Belgium, from 2010 until 2015.**
(DOCX)

**S4 Table. All indicator scores (not controlled for confounders) by year.**
(DOCX)

**S5 Table. The final set of 28 QIs for people with Alzheimer's disease.**
(DOCX)

**S6 Table. The final set of 26 QIs for people with cancer.**
(DOCX)

**S7 Table. The final set of 27 QIs for people with COPD.**
(DOCX)

## Author Contributions

**Conceptualization:** Robrecht De Schreye, Lieven Annemans, Birgit Gielen, Tinne Smets, Joachim Cohen.

**Data curation:** Robrecht De Schreye, Birgit Gielen.

**Formal analysis:** Robrecht De Schreye.

**Funding acquisition:** Lieven Annemans, Birgit Gielen, Tinne Smets, Joachim Cohen.

**Investigation:** Robrecht De Schreye, Joachim Cohen.

**Methodology:** Robrecht De Schreye, Lieven Annemans, Birgit Gielen, Tinne Smets, Joachim Cohen.

**Project administration:** Robrecht De Schreye, Joachim Cohen.

**Resources:** Lieven Annemans.

**Supervision:** Luc Deliens, Tinne Smets, Joachim Cohen.

**Validation:** Luc Deliens, Tinne Smets, Joachim Cohen.

**Visualization:** Robrecht De Schreye.

**Writing – original draft:** Robrecht De Schreye.

**Writing – review & editing:** Robrecht De Schreye, Luc Deliens, Lieven Annemans, Birgit Gielen, Tinne Smets, Joachim Cohen.

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
