## [Decision Letter · Decision Letter 0]

22 Jun 2021

PONE-D-21-14989

Trends in appropriateness of end-of-life care in people with cancer, COPD or with dementia measured with population-level administrative data: do quality indicators improve over time?

PLOS ONE

Dear Dr. Robrecht De Schreye,

Thank you for submitting your manuscript to PLOS ONE. After careful consideration, we feel that it has merit but does not fully meet PLOS ONE’s publication criteria as it currently stands. Therefore, we invite you to submit a revised version of the manuscript that addresses the points raised during the review process.

ACADEMIC EDITOR:

This is a robust well conducted study with important implications for the field of palliative care. There is opportunity to improve the calibre of the reporting in the manuscript.

Please review and respond to the peer reviewer comments.Please rreview and revise the study reporting using guidance for observational studies - STROBE and extension for routine data RECORD. Please include completed checklists as supplementary files, referenced in the methods.Please also proof read carefully to correct all typos.

We look forward to receiving your revised manuscript.

Kind regards,

Catherine J Evans, PhD, MSc, BSc (Hons)

Academic Editor

PLOS ONE

Journal Requirements:

2. In ethics statement in the manuscript and in the online submission form, please provide additional information about the database used in your retrospective study. Specifically, please ensure that you have discussed whether all data were fully anonymized before you accessed them and/or whether the IRB or ethics committee waived the requirement for informed consent. If patients provided informed written consent to have their data used in research, please include this information.

3. In your Data Availability statement, you have not specified where the minimal data set underlying the results described in your manuscript can be found. PLOS defines a study's minimal data set as the underlying data used to reach the conclusions drawn

 in the manuscript and any additional data required to replicate the reported study findings in their entirety. All PLOS journals require that the minimal data set be made fully available. For more information about our data policy, please see http://journals.plos.org/plosone/s/data-availability.

"Upon re-submitting your revised manuscript, please upload your study’s minimal underlying data set as either Supporting Information files or to a stable, public repository and include the relevant URLs, DOIs, or accession numbers within your revised

 cover letter. For a list of acceptable repositories, please see http://journals.plos.org/plosone/s/data-availability#loc-recommended-repositories. Any potentially identifying patient information must be fully anonymized.

Important: If there are ethical or legal restrictions to sharing your data publicly, please explain these restrictions in detail. Please see our guidelines for more information on what we consider unacceptable restrictions to publicly sharing data: http://journals.plos.org/plosone/s/data-availability#loc-unacceptable-data-access-restrictions.

 Note that it is not acceptable for the authors to be the sole named individuals responsible for ensuring data access.

"Funding

This study is part of a research study funded by the Research Foundation Flanders (FWO grant number G012414N) and by the Wetenschappelijk Fonds Willy Gepts."

"Funding for this study was provided by the FWO (Research Foundation - Flanders) and by Wetenschappelijk Fonds Willy Gepts as part of a PhD research project.

7. Your abstract cannot contain citations. Please only include citations in the body text of the manuscript, and ensure that they remain in ascending numerical order on first mention.

8. Your ethics statement should only appear in the Methods section of your manuscript. If your ethics statement is written in any section besides the Methods, please move it to the Methods section and delete it from any other section. Please ensure that your ethics statement is included in your manuscript, as the ethics statement entered into the online submission form will not be published alongside your manuscript. 

9. We note that you have referenced (ie. Bewick et al. [5]) which has currently not yet been accepted for publication. Please remove this from your References and amend this to state in the body of your manuscript: (ie “Bewick et al. [Unpublished]”) as detailed online in our guide for authors

Reviewers' comments:

Reviewer's Responses to Questions

**Comments to the Author**

1. Is the manuscript technically sound, and do the data support the conclusions?

Reviewer #1: Yes

Reviewer #2: Yes

Reviewer #3: Yes

2. Has the statistical analysis been performed appropriately and rigorously? 

Reviewer #1: I Don't Know

Reviewer #2: Yes

Reviewer #3: Yes

3. Have the authors made all data underlying the findings in their manuscript fully available?

Reviewer #1: Yes

Reviewer #2: No

Reviewer #3: Yes

4. Is the manuscript presented in an intelligible fashion and written in standard English?

Reviewer #1: Yes

Reviewer #2: Yes

Reviewer #3: Yes

5. Review Comments to the Author

Reviewer #1: Thank you very much for the opportunity to review this manuscript. The topic is important and the paper strong by its use of linked population-data from eight datasets. My one concern is that 5 years seems a short time period for a trend analysis. That said I enjoyed reading the manuscript and feel it has important findings. Some more specific comments are as follows:

Abstract: The results section is a little wordy - perhaps including some data would improve this. I'm not sure the conclusion necessarily supports the rest of the abstract. It is the first time generalist palliative care, indicators of inappropriate care and medication is mentioned. Perhaps consider re-writing.

Introduction: Clear. Consider including the following relevant reference: Henson LA, Gomes B, Koffman J, Daveson BA, Higginson IJ and Wei G. Factors associated with aggressive end of life cancer care. Supportive Care in Cancer 2016; 24(3):1079-1089.

The reference from Earle et al reports data from the 1990's which is quite out of date now. Consider removing.

Methods: Clear and detailed. Suggest statistical review.

Results: Typo in first sentence with writing of 1ste. I don't find figures 1, 2 and 3 useful. If anything they look like not much has changed over the time period. Consider re-drafting and highlighting a few of the most interesting results rather than a figure with all the findings. Then the scale could be changed so the difference over time more easily visualised.

Discussion: Clear and interesting. Limitations acknowledged.

Reviewer #2: Thank you for the opportunity to review this interesting paper. The article explores an important topic and presents the results of an original research. Understanding changes/shifts in the quality of care provided to patients at the end of life at a population level is key to improve services and, as the ultimate aim, to improve patients’ quality of life. The article is well written, and the methods are clear and rigorous. An important strength of this study is the use of a population-based dataset, of almost the entire population of Belgium. I think some sections and tables could be reported differently to improve clarity.

Find below some specific comments and suggestions by section.

Abstract

Introduction: I think the introduction could be clearer in terms of the importance of measuring the appropriateness of care and make the link between appropriateness as a measure of quality of care, and the importance this can have for improving services.

From my point of view, the emphasis in the conclusion should be put on indicators governments should be focusing on, based on the lack of ‘improvement’ or ‘worsening’, rather than interventions.

Introduction:

In the second paragraph, the sentence ‘In contrast, research shows there has mainly been an increase in indicators…’ reads as if the numbers of indicators have increased. Perhaps it would be clearer if you said there has been an increase in overaggressive care measured through these quality indicators...?

In the third paragraph, you mention the efforts and interest from governments towards measuring the appropriateness/inappropriateness of end-of-life care, but I would suggest giving more emphasis on the relevance of this. Evaluating the quality of care through appropriateness/inappropriateness indicators can help improve service delivery and has the ultimate purpose of improving the quality of life for patients.

Methods

I think you should be clearer (here or in the introduction) about the reasons to include cancer, dementia and COPD patients in the analysis and not other terminal conditions. I agree the three conditions included are important and prevalent and they clearly could benefit from a palliative care approach at the end of life. But there are other conditions too. Does that follow a pragmatical approach in terms of the data available or there is also an argument about the conditions that have been target by the government in Belgium?

It is not very clear why you decided to change the period before death for some indicators to the last 30 days. I can see that approach facilitates comparison between indicators and conditions. But I also assume there was a reason for the period when the indicator was defined. For instance, the appropriateness of a hospital admission might be different if you consider the last 6 months vs the last 30 days.

A table with the list of the different indicators used by condition with their definition would be useful (Perhaps is in the supplementary material?). What an ‘Official palliative care status’ means for instance. It would be easier to follow if you can identify/specify initially which indicators are considered inappropriate or appropriate care. It might not be that intuitive for some readers.

In the ‘Risk adjustment and trend analysis’ section, you mention the variables used as confounders. I think one important variable to consider is the level of multimorbidity. Having multiple comorbidities has been consistently associated with higher use of community health care services and hospital admissions (Browne et al, BMJ Open 2017; Soley-Bori et al, Br J Gen Pract. 2021), and the proportion of adults with two or more comorbidities has increased in the last decade in most European countries. (Souza et al, BMC Public Health. 2021). This is particularly relevant for indicators related to healthcare service use. Is it possible that the variables ‘being entitle to a higher degree of reimbursement due to lower degree of self-reliance’ and ‘being officially recognized as having high care needs’ are representing a measure of multimorbidity or complex healthcare needs in this population? It would be useful to discuss that or give more information about those two variables in order to understand to what extend multimorbidity is a factor that has been accounted for in the analysis.

Results

Typo in the first paragraph of ‘cohort characteristics’ ‘1ste January 2010’ (should be 1st)

In table 1, I would suggest adding columns represent %s to be clear (except for the mean for age). It would be useful to add missing values in this table if present.

In the ‘Risk adjusted differences…’ section, I would suggest using subheading to make it easier to follow. Perhaps by condition or by appropriateness/inappropriateness?

On page 21, please define ‘NOAC’.

In figure 1,2 and 3, I would suggest including a y-axis title. Is the risk-adjusted score what is plotted?

I found the ranking order of the relative increase in table 2 confusing. It is difficult to follow why ICU admissions are at the bottom of the list for COPD and fifth in the list for dementia for instance. I can see that you considered the fact that ICU admissions decreased for dementia patients a ‘positive’ thing and the fact that they increased for COPD patients a ‘negative’ thing. But the column is named ‘relative increase’ and therefore is confusing having + and – changes not in a more ‘intuitive’ order. I would suggest grouping them by ‘appropriateness/inappropriateness’ first, or to be clearer about which indicators are considered appropriate/inappropriate care.

Discussion

In the first sentence, I would say ‘…trends in the appropriateness or inappropriateness of …’ or ‘quality indicators’. Otherwise, it sounds as the increase in the percentage of people admitted to an emergency department is appropriate.

I think you can discuss further the increasing number of contacts with GPs. Several reasons might explain these findings and also they might be influenced by the health care system context. Are there any major changes in health care provision in Belgium for instance that might explain these changes, particularly among dementia patients? An increase in health complexity could also explain higher primary care use. An increase in the number of contacts with GPs is an indirect way of evaluating quality and not necessarily would be considered more ‘appropriate’ in some contexts. It could also be reflecting a lack of coordination or a decrease in the level of continuity of care experienced by patients in primary care. You acknowledge in the limitations the fact that important aspects of the inappropriateness of care are not measured, as this is true not just for contacts with GPs but for all indicators. However, since changes in GP contacts have such an important relative increase, I found it important to discuss it a bit more in this section.

I think it would be important to emphasise in the discussion the fact that the indicators used, although recognised as important to evaluate the quality of care, should be understood as a population-level measure. You would not expect the proportion of people with ED admissions for instance to be reduced to 0, particularly considering end-of-life quality indicators are generally measured retrospectively, and from a clinical point of view it might not be easy to identify when a person is in the last 30 days or the last 180 days before death. That’s another good reason why examining the trend is important, as it gives context to the indicator.

I would be careful with the sentence ‘The increased use of specialized palliative care does not come with a reduction in inappropriate end-of-life care’, as it could be misinterpreted. The measures for each indicator are a population-based measure and therefore, if someone interprets that sentence at an individual level will be falling into an ecological fallacy.

When you say ‘To increase the appropriateness of end-of-life care, palliative care should be initiated earlier…’ it reads to me as if you were suggesting a specific intervention. As I mentioned above, from my point of view the focus of the conclusions should be put on indicators governments should be prioritising, based on their trend, rather than the interventions to do so. There might be different strategies to reduce the number of ED visits for instance.

I think you should give a brief discussion regarding the generalisability of the results.

Reviewer #3: This a great piece of work demonstrating how population-level quality indicators can be used to understood the care provided to people approaching the end of life. It has important policy and practice implications for reviewing trends over time and making decisions for quality improvement and resource allocation. Risk adjusting the trends also demonstrates authors dedication to ensuring that they are not falsely claiming any changes which may be attributable to anything else. It is a very straightforward paper, easy to read accompanied with great figures, and well-written. This paper will be of interest of readers of this journal and a great contribution to the field. The authors can do this huge piece of work more justice by not only focusing on a select few changes that happened over the years, but by also discussing if no change for many others quality indicators is a good thing or not, referring to standards expected at a population level and what it might mean for healthcare and policy makers when a change of a certain magnitude is observed.

Major comments:

1- What's unclear from this paper is what constitutes a "large change". Are you only referring to changes >= 1% as large. this should be clearer in your methods. Additionally, please reflect on why this cut-off is meaningful or significant compared to a change of 0.5% over the years. While I appreciate that the second objective of the study was to identify QIs where the largest changes were observed, where there any QIs which should have improved yet remained stable over the years? It is a shame to include 70 QIs only focus on few, without any reflection or hypothesis around what changes you would expect see to based on the policies etc in place over time.

2- In relation to the comment above, it would be good to remind the readers about the bench-marking values (Maybe in the Supplementary Files?). What is the expected standard for those QIs. This would make interpretations of the trend for each QI more meaningful. I appreciate that these values might be difficult to determine yet difficult for a reader to understand where things may need improvement without a benchmark.

3- Discussion, especially the interpretation section of this manuscript seems a bit rushed. It focuses mainly on people with dementia and simply brushes over findings regarding the care of people with COPD and cancer. Please revise this section to reflect both overall and disease-specific interpretations of the findings.

4- Interpretation para 2 - Could it be also because of palliative care involvement, things which may have not been picked up previously are noticed and hence people end up in the ED? People with dementia have complex care needs. How can we be sure that all ED visits are inappropriate? If there are no community alternatives to replace the care provided in the ED, what should happen? If there are alternatives providing very similar services in Belgium, please reflect on this.

5- Interpretation para 3 - Some barriers to de-prescription are stated. Are these barriers faced even after using a review tool such as STOPPFrail? Please reflect on further implications of polypharmacy, multimorbidities and ED attendances. These are likely to be interconnected rather than separate.

6- Implications - "to increase appropriateness of end-of-life care, an effort should be made to reduce the number of ED and ICU admissions of people with serious and chronic illness near the end of life. Agreed that this is important but why are people having these admissions? Surely, this is a research question for future research/direction. Also this is oversimplyfing the situation and saying that if we reduce these admissions, we will increase the provision of appropriateness of EOL care.

7- Authors should acknowledge that it is difficult to understand whether the care was appropriate or not by simply using QIs. QIs are very useful for understanding population-level care and their use here is correct, but this is a limitation.

8- There is some mention of policy and attitude shift in Belgium around the importance of palliative care and a statement around an improvement in trends might be expected. Bringing in more context here and in the Discussion might be helpful. This would also make it easier to understand the findings' implications on future policy and quality improvement.

Minor comments:

1- There are few typos here and there (health care and healthcare, 1ste Jan etc.) - please have a look at these.

2- Your title can be simplified to "Trends in appropriateness of end-of-life care in people with cancer, COPD or with dementia measured with population-level quality indicators". This is just a suggestion because "Do quality indicators improve over time" does not really make sense. "Does quality of care improve over time" makes more sense, but one QIs are only one part of measuring and understanding the quality of care.

3- Introduction: "These efforts in research and policy might be a reflection of a broader societal evolution towards more attention for appropriateness of end-of-life care and positive attitudes towards palliative care." Please back this up with a reference.

4- The first objective of the study reads as not a trend analysis of change over years but a comparison between 2010 and 2015. The second objective sounds grammatically incorrect. Could you please re-word these?

5- Could you please add the percentages for "being officially recognized as having high care needs" and "being entitled to a higher degree of reimbursement due to lower degree of self-reliance" in the descriptive tables as they were also entered into regression model for risk adjustment?

6- Please add to your tables that you are reporting percentages.

7- In the results section, when referring to disease specific trends, please only refer to the relevant figure.

8- Table 2 - Think carefully about what you'd like the readers to get from this table. It is a busy and hard to read table, which actually contains valuable information. If you would like readers to evaluate each disease by themselves, please list the improvement scores for each disease group one after other rather than side by side. If you would like readers to easily compare some of the same QIs across disease (e.g. ED admissions, palliative care involvement), please list the QIs common for all groups first, then the others side by side. Either way, this table needs to be revised.

9- The first limitation is about relying on death certificate data to select the COPD and dementia populations but there is no references or explanations about the COPD population's identification.

10- Interpretation Para 2 - People with dementia near the end of life in Belgium often live in a nursing home. - Please provide a reference or give the percentage from your findings.

11- Figures - Please use the x-axis used in Figure 2, INDICATORS OF INAPPROPRIATE GENERAL TREATMENT AND MEDICATION. for all other figures as this is the most readable one.

12- Supplementary Files - Typo - secundary and please state for net taxable income quintiles 1-5 which one indicates the highest vs lowest.

6. PLOS authors have the option to publish the peer review history of their article (what does this mean?). If published, this will include your full peer review and any attached files.

Reviewer #1: No

Reviewer #2: No

Reviewer #3: No

---

## [Author Response · Author response to Decision Letter 0]

19 Jan 2022

Dear editor, 

We submitted a file named "response to reviewers", containing all reviewers comments and our response to them, with changes to the manuscript where needed. 

Kind regards, 

Robrecht De Schreye, corresponding author

---

## [Editor Report · Decision Letter 1]

14 Mar 2022

PONE-D-21-14989R1Trends in appropriateness of end-of-life care in people with cancer, COPD or with dementia measured with population-level quality indicatorsPLOS ONE

Dear Dr. Robrecht De Schreye,

Thank you for submitting your manuscript to PLOS ONE. After careful consideration, we feel that it has merit but does not fully meet PLOS ONE’s publication criteria as it currently stands. Therefore, we invite you to submit a revised version of the manuscript that addresses the points raised during the review process. Thank you for careful consideration of the peer review and editor comments. You have addressed these well in the manuscript and strengthened greatly the clarity and quality of your reporting. I have indicated minor revisions. This pertains to the abstract and few minor points:

1. Abstract - the results and conclusions could be strengthened. You have important novel findings that could be better conveyed in the abstract. You need to reduce words in the preceding sections to increase reporting in the results. Reduce words in the intro - this is too long. Remove the years from the aim, this is stated in the design, and just state Design: A population-level ....Participants: All people who died.... Removing we conducted - it is not needed. Results - We identified three main trends overtime across the three disease groups of increasing use of: family physician in the last 30 days of life (give +% for each group); receipt of specialist palliative care mostly initiated in the last 14 days of life (give +% for each ; and use of emergency department in the last 30 days of life (give +% for each). Remove sentence re additional trends as not giving any information - or keep and give examples of these trends

Conclusions - don't really link to your results. Your conclusion in the manuscript is much stronger and clearer. Can you review and revise detailing e.g. identification of appropriate care xx and inappropriate to answer your stated aim, then focus for policy makers.

2. Minor points -

• Introduction - Earle et al 2008 reference reporting data > 30 years old. I can see that important to have reference re use of chemo, but there must be a more recent reference given high income countries typically monitor use of chemo in last 30 days of life. Can you review and update. it is a bit distracting to use such old data when you are arguing about more recent change

• Use of language - terms of handicap and elderly people are not generally used as considered derogatory. Can you remove - this is line 245, and state judged by the GP for a person with multiple complex needs - you could give examples polypharmy, high level of disability, multimorbidity. FYI see useful publication on language - Lundebjerg NE, Trucil DE, Hammond EC, Applegate WB. When It Comes to Older Adults, Language Matters: Journal of the American Geriatrics Society Adopts Modified American Medical Association Style. Journal of the American Geriatrics Society 2017; 65(7): 1386-8.

• Results - table 1 - state Age mean (SD) - you need to report the (SD) for each mean given - as standard reporting 68.2 (insert SD). This is important for the interpretation of the data presented

We look forward to receiving your revised manuscript.

Kind regards,

Catherine J Evans, PhD, MSc, BSc (Hons)

Academic Editor

PLOS ONE

Journal Requirements:

Additional Editor Comments (if provided):

See above comments

Reviewers' comments:

No further reviewer comments. 

---

## [Author Response · Author response to Decision Letter 1]

27 Apr 2022

Dear editor,

Thank you for your careful consideration of our submitted manuscript. We appreciate your feedback and we fully agree your suggestions improve the quality of the article. We integrated your suggestions into the manuscript.

We hope you find our adjustments acceptable and we are looking forward to hearing from you. 

We confirm that this manuscript has not been published elsewhere and is not under consideration by another journal. All authors have approved the manuscript and agree with its submission.

With my best regards,

Robrecht De Schreye, PhD

on behalf of all co-authors

---

## [Decision Letter · Decision Letter 2]

29 Jun 2022

PONE-D-21-14989R2Trends in appropriateness of end-of-life care in people with cancer, COPD or with dementia measured with population-level quality indicatorsPLOS ONE

Dear Dr. Schreye,

Thank you for submitting your manuscript to PLOS ONE. After careful consideration, we feel that it has merit but does not fully meet PLOS ONE’s publication criteria as it currently stands. Therefore, we invite you to submit a revised version of the manuscript that addresses the points raised during the review process.

Thank you for the opportunity to review the manuscript titled “Trends in appropriateness of end-of-life care in people with cancer, COPD or with dementia measured with population-level administrative data: do quality indicators improve over time?” The authors observed mixed trends of end-of-life care in people dying from cancer, COPD or with dementia using indicators of appropriateness and inappropriateness of end-of-life care. This manuscript is well written. However, I have a few minor concerns about the methods and would like some clarifications.

Minor:

1. Introduction: the authors mentioned that “Earle et al., for instance, found an increase in chemotherapy use and continuation of chemotherapy closer to death in Medicare patients with cancer in the US between 1993 and 1996.10 ” However this evidence applies to the trend in chemotherapy more than 20 years ago. I would suggest that the authors should use more recent references.

2. Methods: It seems the analytic cohort only consists of the decedents whose health care use was reimbursed. Despite the authors acknowledged the limitation in the paper, it would be helpful to describe the proportion of reimbursed care and non-reimbursement care, so that readers can have a general sense about how representative the analytic sample is.

3. Method: The risk-adjusted score was calculated for each year. However, only the risk adjusted scores of 2010 and 2015 were used to calculate the increase and decrease. This is certainly not trend analysis, with the information between these two years all missing. I would suggest a trend test by incorporating the adjusted indicator scores of year 2011-2014.

4. Discussion: the authors observed an increase in palliative care while an increase in ED visits and ICU admission too. The discussion about the contradicting trends should be strengthened, i.e. what is the possible reason? Is there any policy change during the study period?

We look forward to receiving your revised manuscript.

Kind regards,

Lihua Li, Ph.D.

Academic Editor

PLOS ONE

Journal Requirements:

Reviewers' comments:

Reviewer's Responses to Questions

**Comments to the Author**

1. If the authors have adequately addressed your comments raised in a previous round of review and you feel that this manuscript is now acceptable for publication, you may indicate that here to bypass the “Comments to the Author” section, enter your conflict of interest statement in the “Confidential to Editor” section, and submit your "Accept" recommendation.

Reviewer #2: All comments have been addressed

Reviewer #3: All comments have been addressed

2. Is the manuscript technically sound, and do the data support the conclusions?

Reviewer #2: Yes

Reviewer #3: Yes

3. Has the statistical analysis been performed appropriately and rigorously? 

Reviewer #2: Yes

Reviewer #3: Yes

4. Have the authors made all data underlying the findings in their manuscript fully available?

Reviewer #2: Yes

Reviewer #3: Yes

5. Is the manuscript presented in an intelligible fashion and written in standard English?

Reviewer #2: Yes

Reviewer #3: Yes

6. Review Comments to the Author

Reviewer #2: Thank you for the opportunity to review the revised version of this manuscript. I believe the authors have thoroughly my comments and incorporated useful changes to the manuscript. I do not have further comments.

Reviewer #3: The authors did address all the comments from the previous review. This will be very valuable contribution to the field (both in terms of knowledge and methodology). I made some suggestions to the language and minor suggestions for the discussion.

Abstract aim could be reworded for clarity - "To evaluate the appropriateness of end-of-life care for people with cancer, COPD or dementia in Belgium, using quality indicators with routinely collected population level data"

Introduction - Line 84, Double negative could be confusing "negative impacts outweigh the benefits in terms of QoL"

Discussion, Main Findings - Please add increase in ICU use here as well

Interpretation, Line 432 - Could you please reflect on what could be the additional benefit on early initiation of specialist pall care involvement?

7. PLOS authors have the option to publish the peer review history of their article (what does this mean?). If published, this will include your full peer review and any attached files.

Reviewer #2: No

Reviewer #3: No

---

## [Author Response · Author response to Decision Letter 2]

12 Aug 2022

Thank you for taking the time to assess our manuscript. 

Minor: 

1. Apparently something went wrong with the version that was provided to the editor. In the current version, the text the editor refers to is no longer included in the manuscript. In the last response to the reviewers, following a similar question, we updated the reference with more recent literature and changed the text accordingly.

2. Subscription to health insurance is legally mandatory in Belgium for all citizens. As such, the cohort consists of almost the full population. This is mentioned on lines 142-144 of the methods section. 

The limitation mentioned in the discussion is about types of care that are not reimbursed, rather than people not being included in the study. This is explained in lines 427-430 of the discussion.

3. We agree with the editor: the calculation of the score differences between 2010 and 2015 was an addition to and not part of the trend analysis. 

This is described in lines 261 – 270 of the methods section. We aimed to make the distinction by writing: “This way, a trend analysis was performed, (…).We also calculated all risk adjusted indicator score differences between 2010 and 2015.” We hope this clarifies these are two distinct steps. 

The results of the trend analysis are depicted in figures 1, 2 and 3, while the results of the difference calculation is shown in table 2. 

4. During the study period, there were no policy changes that could explain the increases. We agree with the editor this should be made more clear in the text. 

We also agree the argument by Teno et al, corroborated by our findings, should be presented more strongly. 

We changed the text accordingly.

---

## [Editor Report · Decision Letter 3]

22 Aug 2022

Trends in appropriateness of end-of-life care in people with cancer, COPD or with dementia measured with population-level quality indicators

PONE-D-21-14989R3

Dear Dr. Schreye,

We’re pleased to inform you that your manuscript has been judged scientifically suitable for publication and will be formally accepted for publication once it meets all outstanding technical requirements.

Kind regards,

Lihua Li, Ph.D.

Academic Editor

PLOS ONE